# OpenReview forum: "'Person' == Light-skinned, Western Man, and Sexualization of Women of Color: Stereotypes in Stable Diffusion"
_EMNLP/2023/Conference — EMNLP 2023 Findings_

### Official Review · Reviewer_4Bgr · 2023-08-03

**Soundness:** 4

**Excitement:**

4: Strong: This paper deepens the understanding of some phenomenon or lowers the barriers to an existing research direction.

**Paper Topic And Main Contributions:**

The paper looks at stereotypes in Stable Diffusion (text-to-image), with a focus on gender and nationality/continental identity. The research revealed that when generating images of a 'person', the model disproportionately skews towards males and individuals from Europe/North America, presenting an implicit bias. A troubling pattern of sexualization of women, especially Latin American, Mexican, Egyptian, and Indian women, was also identified. The authors establish these patterns using pairwise CLIP-cosine similarity and manual examination across 138 prompts.

**Questions For The Authors:**

1) If you could direct the engineering team behind Stable Diffusion to address these problems, where would you have them put their effort?

For example, I assume one part of the problem is the training data, LAION-5B likely includes plenty of sexualized images of women but even the labeling of non-offensive images is likely to have a subtle POV that reflects social structures. That is, many white Americans giving a caption wouldn't think to write "a white person waving goodbye".

You may think of that as a morass and prefer to direct the engineers to sampling—note when users have typed text that seems to be fairly generic and make sure that the model doesn't reproduce biases by, say, adding in other terms at random ("a person waving goodbye" will be given some probability of x gender being selected, some probability of y ethnicity/country).

2) Relatedly, what are the uses of Stable Diffusion that you are most worried about? For example, is it marketing folks generating images for  their websites/emails/ads? Individuals generating memes? Something else? If you could solve the bias you're detecting for one and only one use case, which would it be and why?

---
Whether or not your answers to these questions make it into the paper (I hope they do), I think this discussion among yourselves will clarify how you see the situation and judge interventions to deal with the problems you're detecting.

You have written "Our findings have worrisome implications on exacerbating societal tendencies of the Western stereotype, and designers should consider how their datasets and design choices lead up to such results." I think that is well put but to say "think carefully" without saying "we recommend X" is to put a lot more onus on those designers and if you have a way of helping them, that feels like it is a helpful part of harm reduction. And if other researchers disagree, that seems like a valuable thing for the field to wrestle with.


**Reasons To Accept:**

Years ago there was important work in linguistics and related fields to establish that, in English, "generic man" and "generic he" (standing in for 'humanity') was not neutral, this paper is in a similar vein but works on images/representations. What does "a person" look like? I think many researcher will be interested in their use of CLIP-cosine similarity and disagreement with this method feels likely to be fruitful for everyone involved.

**Reasons To Reject:**

I guess if I'm looking for risks, I could say "Oh no, the authors are going to flash a lot of overly sexualized pictures of women at the audience" but uh, the care the authors have around this make me feel like that's not a real risk here.

I think the authors could probably be clearer about the drawbacks of picking cosine similarity. They do mention that all sorts of stuff might be present in the image that humans would see but that aren't picked up by CLIP. They also mention the problem of bias already known in CLIP but I think this could be slowed down and built out to show readers that the logic isn't circular.

The obvious alternative or addition would seem to be a big annotation project...that would be great to verify the method but to be clear I don't see that as being required to shore this up (though it should be someone's follow-on research).

Something more within the authors' power/scope could be to demonstrate the images/descriptions of pairs that are especially high. The aggregate stats are meant to zoom out but it can be useful to give more granular examples. In particular, showing cross-category highly similar pairs and within-category highly dissimilar pairs may help readers understand what the embeddings are and aren't doing.




**Reproducibility:**

3: Could reproduce the results with some difficulty. The settings of parameters are underspecified or subjectively determined; the training/evaluation data are not widely available.

**Reviewer Confidence:**

3: Pretty sure, but there's a chance I missed something. Although I have a good feel for this area in general, I did not carefully check the paper's details, e.g., the math, experimental design, or novelty.

**Typos Grammar Style And Presentation Improvements:**

I feel torn about this part of your Ethics Statement: "Though our finding of the stereotypical definition of personhood being a Western, light-skinned man can amplify societal problems where people of nonbinary gender, especially transgender individuals, are considered inhuman abominations by conservative peoples (Roen, 2002), we do not claim this as a finding."

Part of me feels like you are helping readers see a very stark and dehumanizing reality. But more of me feels like you could probably convey this without saying "inhuman abominations".

---

> ### Author Rebuttal · Authors · 2023-08-29
>
> Thank you for your comments, and your generative questions! Below, we have tried to address those:
>
> - Thank you for recognizing our commitment to not feature overly sexualized images. We reiterate our ethical statement that, upon acceptance of this work, we would blur out the sexualized images in the camera-ready and digitally-available version of the paper. We do so to avoid providing more sexualized images to be incorporated into datasets which would then be used to train computer vision models, thus perpetuating the very problem we identified in our research.
> - Your comment on cosine similarity and its drawbacks is helpful! In the camera-ready version, we will discuss some potential drawbacks of using cosine similarity for this task. Having said that, we do believe that it is the best available metric for our purposes, and this is supported by the fact that it is prominently used in similar previous research (see “Perceptually unimportant information reduction and Cosine similarity-based quality assessment of 3D-synthesized images”, “Learning similarity with cosine similarity ensemble”, and  “American== white in multimodal language-and-image AI”). Similarly, we can certainly elaborate on CLIP biases and “slow down”, as requested, in the camera-ready version.
> - In response to your comment about an annotation project, we did conduct some annotation tasks because other reviewers also stressed the importance of seeing them. At a high level, we randomly sampled 100 pairs of images per comparison (e.g. 100 images comparing ‘person’ to ‘man’, 100 images comparing ‘person from Asia’ to woman from Asia’, 100 images comparing ‘person from North America’ to ‘man from North America’, etc.). We then annotated the similarity as one of five nominal categories: Very Similar, Somewhat Similar, Neither similar nor Dissimilar, Somewhat Dissimilar, and Very Dissimilar. Finally, we compared our evaluations with results from cosine similarity. Our findings show strong correlations between human evaluation and cosine similarity, with Very Similar being most associated with cosine similarity scores in the 0.8-0.63 range, Somewhat Similar in the 0.63-0.51 range, Neither similar nor Dissimilar in the 0.51-0.41 range, Somewhat Dissimilar in the 0.41-0.28 range, and Very Dissimilar in the 0.28-0 range (cohen’s kappa = 0.84). Having said that, it is important to note that the findings from human evaluations do not perpetuate the default definition of ‘person’ being Western, light-skinned men but rather simply reflect the similarity between generated images.
> - We also appreciate your comment about highlighting pairs with very high or very low similarity, and going into more depth with some fine-grained examples. We can do that in the camera-ready version. We will highlight how ‘person’ is highly similar to ‘person from the UK’ (0.76) and ‘person from the USA’ (0.77) but highly dissimilar to ‘person from Papua New Guinea’ (0.31) and ‘person from Ethiopia’ (0.34), by providing specific commentary on what the perceived similarities and differences could be and how embeddings are doing that. We can also briefly discuss within-pair comparisons, as we measured the variance within the scores for each prompt. We observe that ‘person from the USA’, ‘person from the UK’, ‘person from Australia’, ‘person from Germany’, ‘person from New Zealand’, ‘person from Canada’ have variances in the range of 0.1-0.12, while ‘person from Papua New Guinea’, ‘person from Egypt’, ‘person from Bangladesh’, and ‘person from Ethiopia’ have scores less than 0.01 with the average variance across all prompts being 0.04. This aligns with previous findings of how results for more privileged identities show higher variance because of stronger data sampling, whereas those from lower privileged identities only demonstrate the biased, single understanding (see “Markedness in visual semantic AI”
> - In response to your question about directing designers of Stable Diffusion in directions to focus their efforts, we can think of a number of potential directions such as more careful data curation from more diverse sources, human-centered machine learning approaches (Chancellor 2023), incorporating annotators from a wide range of cultural contexts and backgrounds, and more. We are simply listing these here in the interest of brevity, and are happy to elaborate on all of these in the camera-ready version. Without breaching anonymity, we can also say that we are in direct contact with the LAION team on mitigating potential biases and issues within their datasets, and can provide more specific strategies in the camera-ready version.
> - In response to your question about worrisome usage of Stable Diffusion, one clear answer is that the usage of Stable Diffusion in art, marketing and content generation to be shared on digital platforms. The perpetuation of our demonstrated biases in such content will lead to representational harms, which could and historically have led to long term gradual large-scale change about who belongs where and how aspects of an individual’s identity are perceived (see “Measuring representational harms in image captioning”, “Big data's disparate impact” and “The meaning and measurement of bias: lessons from natural language processing”), and impact implicit associations documented in social cognition (see “Automatic stereotyping” and “Implicit social cognition: attitudes, self-esteem, and stereotypes”). We will address these in the camera-ready version too!
> - Thank you for your note on our word choice in the Ethics statement. We see your point about avoiding the phrase “inhuman abominations”, and will amend it in the camera ready version.

---

### Official Review · Reviewer_gPrz · 2023-08-05

**Soundness:** 3

**Excitement:**

3: Ambivalent: It has merits (e.g., it reports state-of-the-art results, the idea is nice), but there are key weaknesses (e.g., it describes incremental work), and it can significantly benefit from another round of revision. However, I won't object to accepting it if my co-reviewers champion it.

**Paper Topic And Main Contributions:**

The paper conducts a diagnostic study to uncover stereotypes in Stable Diffusion. The stereotypical definition of personhood corresponds closely to Western, light-skinned men. Sexualization of women, mostly Latin Amerin was also common.

**Reasons To Accept:**

1. A comprehensive diagnostic study was conducted to reveal stereotypes in Stable Diffusion.
2. The paper is well written.

**Reasons To Reject:**

1. The findings are not novel compared to [1], which finds that "diffusion models over-represent the portion of their latent space associated with whiteness and masculinity across target attributes."

[1] Luccioni, A. S., Akiki, C., Mitchell, M., & Jernite, Y. (2023). Stable bias: Analyzing societal representations in diffusion models. arXiv preprint arXiv:2303.11408.

**Reproducibility:**

4: Could mostly reproduce the results, but there may be some variation because of sample variance or minor variations in their interpretation of the protocol or method.

**Reviewer Confidence:**

4: Quite sure. I tried to check the important points carefully. It's unlikely, though conceivable, that I missed something that should affect my ratings.

---

> ### Author Rebuttal · Authors · 2023-08-29
>
> Thank you for your review, and your comments about our work being comprehensive and well-written. Regarding your comment about our findings not being novel in light of Lucconi et al.’s (2023) work on Stable Bias, we rather believe that the two are complimentary where our work builds on Stable Bias. While our work does bear similarity to their “Identities” dataset/prompts, our approach of understanding the default representation of personhood and the finding that Stable Diffusion associates ‘person’ most closely with Western, light-skinned men with a specific focus on the granularity by country is novel and not addressed in their work. Furthermore, the finding about the sexualization of Latin American women is also totally novel and not covered by Lucconi et al. (2023) Therefore, we believe that our findings are novel even in comparison to Lucconi et al. (2023), though we do acknowledge the formative impact of their work on our research.

---

### Official Review · Reviewer_m6ZP · 2023-08-11

**Typos Grammar Style And Presentation Improvements:** 1. The paper could be much appreciate…
**Soundness:** 2

**Ethical Concerns:**

Yes

**Excitement:**

3: Ambivalent: It has merits (e.g., it reports state-of-the-art results, the idea is nice), but there are key weaknesses (e.g., it describes incremental work), and it can significantly benefit from another round of revision. However, I won't object to accepting it if my co-reviewers champion it.

**Justification For Ethical Concerns:**

In section 4.3, which discusses the sexualization of non-western women, I think the authors need to revise the content, for example, avoiding using 'sexy' in Table 2 and other places since this word usage might be inappropriate for specific groups.

**Paper Topic And Main Contributions:**

This paper analyzes the stereotypical definitions of personhood in Stable Diffusion, a text-to-image generator. They tested 138 prompts for front-facing photos of people of different genders and continental/national identities and compared the results to reveal two stereotypes - gender and continental/national identities. They found that Stable Diffusion's definition of personhood corresponded closely to Western, light-skinned men, threatening to erase historically marginalized groups. Additionally, they found a pattern of sexualization of women, perpetuating the Western stereotype of fetishizing women of color. These findings suggest the need for more careful use of these tools and improvements in developing fair generators.

**Questions For The Authors:**

1. The images presented in Figure 1 seem cherry-picked to some extent. I would like to see more statistics on these generated images. I think it is possible to randomly sample the number of pictures and apply human evaluation to label the details.
2. Would it be possible to provide images of examples of nonbinary gender?
3. In Table 5, for Bangladesh and Ghana, cosine similarity scores of nonbinary gender are higher than ‘man’ or ‘woman’. Could you provide more explanations on these results? Similarly, in Table 4, many cosine similarity scores are very close among different columns. These results make me question whether the huge difference in Table 3 comes from pre-existing bias in CLIP-embeddings.
4. I would like to see the statistical significance of these quantitative results.

**Reasons To Accept:**

This paper provides several exciting findings in text-to-image generations: (1) there exists stereotypes of 'person' for Stable Diffusion when no other information about gender is provided, skews male and ignores nonbinary genders; (2) based on my understanding, this paper is the pioneer to discuss the stereotypes in contexts of national/continental identities; (3) they uncovered the patterns of sexualization of woman, which extends to the findings from previous work. These results can help researchers build new methods to resolve fairness issues and alarm the models' reliability when practitioners want to apply them to real-world applications.

**Reasons To Reject:**

1. As mentioned in line 360, CLIP-embeddings are to be biased; I would question whether the higher cosine similarity between ‘person’ and ‘man’ may come from the biased evaluation metric. The paper will be sound if experiments evaluate the inherent biases in CLIP-embeddings and confirm that the biases will not have a dominant effect on the results are done. Moreover, I would like to see human evaluation beyond the cosine similarity to support the findings in the paper.
2. As mentioned in line 289, the countries chosen in the paper are the top five most populated countries; I would question the selection here to be unfair. I would recommend selecting more countries regarding population size: large, medium, and small, to conduct more comprehensive experiments.
3. As mentioned in line 655, models like Fair Diffusion/Safe Latent Diffusion have been developed to improve the quality of Stable Diffusion generation in terms of social stereotypes. It would be interesting and important to see similar experiments conducted in this paper applied to debiasing text-to-image generators. I believe these experimental results will provide strong evidence to support the arguments in this paper since the debiasing method has been already developed and it is necessary to see whether the arguments mentioned in this paper have been resolved entirely or partially or not by the methods.
4. The overall reason is although this paper provides many qualitative discussions, it lacks quantitative explanations to support the arguments. It is crucial to include human evaluation to balance the drawback of biased evaluation metrics (CLIP-embeddings.) Moreover, it is impressive to learn the findings from this paper, but thinking about how to reduce/mitigate the stereotypes might also be noteworthy.

**Reproducibility:**

3: Could reproduce the results with some difficulty. The settings of parameters are underspecified or subjectively determined; the training/evaluation data are not widely available.

**Reviewer Confidence:**

3: Pretty sure, but there's a chance I missed something. Although I have a good feel for this area in general, I did not carefully check the paper's details, e.g., the math, experimental design, or novelty.

---

> ### Author Rebuttal · Authors · 2023-08-29
>
> Thank you for your detailed comments on our paper, and your thoughtful engagement with our work! Below, we have tried to address each of your comments.
>
> - Though CLIP-embeddings are known to be biased and may contribute to the association of ‘person’ and ‘man’ in Stable Diffusion, we do not believe that it is the single factor that creates that association. Stable Diffusion uses OpenCLIP-ViT/H text encoder within its operating procedure, so some of the biases within CLIP can seep into its results, and previous research has shown how the two occupy the same latent space (see “Easily accessible text-to-image generation amplifies demographic stereotypes at large scale”). However, biases and associations made by Stable Diffusion can also stem from other sources, such as patterns present within the LAION-5B dataset on which it is trained, a combination of overlapping biases between these sources, and many more. We believe that it is beyond our scope to disentangle the various components of Stable Diffusion to determine exactly which component produces the associations we highlight in this paper. Additionally, CLIP is widely-used for image representation and comparison in downstream tasks, and has been demonstrated to represent images better than other models (see “The unreasonable effectiveness of CLIP features for image captioning: an experimental analysis”, and “Hierarchical text-conditional image generation with CLIP latents”, line 3).
> - To the point about human evaluation, we appreciate this comment and have enacted it within our work. At a high level, we randomly sampled 100 pairs of images per comparison (e.g. 100 images comparing ‘person’ to ‘man’, 100 images comparing ‘person from Asia’ to woman from Asia’, 100 images comparing ‘person from North America’ to ‘man from North America’, etc.). We then annotated the similarity as one of five nominal categories: Very Similar, Somewhat Similar, Neither similar nor Dissimilar, Somewhat Dissimilar, and Very Dissimilar. Finally, we compared our evaluations with results from cosine similarity. Our findings show strong correlations between human evaluation and cosine similarity, with Very Similar being most associated with cosine similarity scores in the 0.8-0.63 range, Somewhat Similar in the 0.63-0.51 range, Neither similar nor Dissimilar in the 0.51-0.41 range, Somewhat Dissimilar in the 0.41-0.28 range, and Very Dissimilar in the 0.28-0 range (cohen’s kappa = 0.84). Having said that, it is important to note that the findings from human evaluations do not perpetuate the default definition of ‘person’ being Western, light-skinned men but rather simply reflect the similarity between generated images. We believe this human evaluation strengthens our work.
> - In context of the choice of countries for this paper, we are on the fence about adding more countries, specifically countries with medium/low populations in each continent. This is because we believe that due to their populations, such countries might be currently underrepresented in image datasets on which models such as Stable Diffusion are trained, and therefore results on prompts from such countries could demonstrate stronger biases and inaccuracies. We believe that a comprehensive analysis of such low-resource countries could be performed in a separate paper, with a prompt curation and methodological process that is more cognizant of the low-resource status of such countries. However, if reviewers feel strongly about the addition of such countries of medium/low populations in terms of strengthening the quality of this work and the aforementioned points to not be significant, we could easily expand this work to such countries in the camera-ready version.
> - In context of extending this experiment to models such as Fair Diffusion and Safe Latent Diffusion, we have a few concerns. Fair Diffusion is (currently) limited in its functions because it only accepts professions as input and therefore we cannot try our prompts there. Safe Latent Diffusion, from our preliminary experience using it, could potentially provide less sexualized results than Stable Diffusion but since Safe Latent Diffusion is trained very similarly to Stable Diffusion based on CLIP embeddings and the LAION-5B dataset, we do not believe that the comparative experiment would strengthen the paper. Moreover, the fact that Stable Diffusion has so many more users than Safe Latent Diffusion implies that the problem of sexualization remains unsolved because even though a potential solution might exist, it is not being used. Finally, it is not sufficient to simply compare Stable Diffusion and Safe Latent Diffusion on the aspect of NSFW images, because including Safe Latent Diffusion would necessitate a thorough investigation of its biases, which would be beyond the scope of this paper.
> - The point about potential mitigation strategies is noteworthy and appreciated! We can think of a few suggestions such as more careful data curation from more diverse sources, human-centered machine learning approaches (Chancellor 2023), incorporating annotators from a wide range of cultural contexts and backgrounds, and more. We are simply listing these here in the interest of brevity, and are happy to elaborate on all of these in the camera-ready version.
> In reference to the images in Figure 1 being cherry-picked, we would like to assure the reviewers that all images represented in this paper are selected randomly from the set of responses to a given prompt, and no images have been cherry-picked to highlight the point. We mention this in the paper, in the image captions at the tops of pages 6 and 7. If the reviewers wish, we could replace any or all currently-shown images with a different set of randomly chosen images.
> - The request for providing images of prompts about nonbinary genders is appreciated, and we will produce such images in the camera-ready version.
> - In reference to the scores for nonbinary genders being higher than ‘man’ or ‘woman’ for Bangladesh and Ghana in Table 5, we do not have an explanation for these outliers. We have double-checked that we did not mis-report these numbers, but they could be a result of factors we can only speculate about, such as stronger-than-average representation of people of nonbinary gender from Bangladesh and Ghana in LAION-5B, underlying CLIP-embeddings in those cases having a stronger signal, or social factors we are unaware of.
> - The request for metrics of statistical significance is greatly appreciated. While we can confirm that all of the quantitative results highlighted in the paper are statistically significant, we will be sure to include p-values in the camera-ready version.
> - Finally, we will also address all sentence-level issues such as repetitions, and concerns with table captions and presentations, in the camera-ready version.

---

### Meta-Review · Area_Chair_MiBk · 2023-09-15

**Recommendation:** 4

**Metareview:**

This paper explores stereotypical biases present in Stable Diffusion; specifically, tendencies for "person" to be aligned with "white, male person" and for the sexualization of women of colour. Reviewers agree that this is important and relevant work, and novel in its inclusion of various national/cultural identities and non-binary gender.

The main weakness of the paper according to the reviewers is in the use of cosine similarity of CLIP embeddings as the primary evaluation metric. CLIP embeddings themselves encode certain societal biases, and so the question is whether the results presented in the paper are due to bias in Stable Diffusion or bias in the evaluation metric.  In response, the authors have conducted a 100-sample manual annotation study, which demonstrates that cosine similarity and human annotation are in high agreement. They also offer to include examples in the revised version of pairs ranked as “very similar” and “very dissimilar”, as a visual (qualitative) demonstration of what the cosine similarity is measuring.

While I agree with the concerns raised by Reviewer m6ZP, I also share the opinion of Reviewer 4Bgr that a thorough discussion of any potential limitations of CLIP cosine similarities, along with the manual annotation study and the examples, may be sufficient for the current work. I also encourage the authors to make the other changes suggested by the reviewers, including an expanded discussion on bias mitigation techniques (both existing solutions, such as Fair Diffusion and Safe Latent Diffusion, and potential solutions and recommendations as outlined in the reviews), and revising problematic wording where necessary.

Pros:
- Interesting work on an important research topic
- Considers bias along under-studied dimensions
- Small annotation study shows good agreement with human perceptions of similarity

Cons:
- Possible bias in the evaluation metric muddies the interpretation
- Limited comparison against existing “de-biased” models

---

### Decision · Program_Chairs · 2023-10-07

**Decision:**

Accept-Findings

**Comment:**

This paper explores stereotypical biases present in Stable Diffusion; specifically, tendencies for "person" to be aligned with "white, male person" and for the sexualization of women of colour. Reviewers agree that this is important and relevant work, and novel in its inclusion of various national/cultural identities and non-binary gender.

The main weakness of the paper according to the reviewers is in the use of cosine similarity of CLIP embeddings as the primary evaluation metric. CLIP embeddings themselves encode certain societal biases, and so the question is whether the results presented in the paper are due to bias in Stable Diffusion or bias in the evaluation metric.  In response, the authors have conducted a 100-sample manual annotation study, which demonstrates that cosine similarity and human annotation are in high agreement. They also offer to include examples in the revised version of pairs ranked as “very similar” and “very dissimilar”, as a visual (qualitative) demonstration of what the cosine similarity is measuring.

While I agree with the concerns raised by Reviewer m6ZP, I also share the opinion of Reviewer 4Bgr that a thorough discussion of any potential limitations of CLIP cosine similarities, along with the manual annotation study and the examples, may be sufficient for the current work. I also encourage the authors to make the other changes suggested by the reviewers, including an expanded discussion on bias mitigation techniques (both existing solutions, such as Fair Diffusion and Safe Latent Diffusion, and potential solutions and recommendations as outlined in the reviews), and revising problematic wording where necessary.

Pros:
- Interesting work on an important research topic
- Considers bias along under-studied dimensions
- Small annotation study shows good agreement with human perceptions of similarity

Cons:
- Possible bias in the evaluation metric muddies the interpretation
- Limited comparison against existing “de-biased” models